# Research on High-Precision Resonant Capacitance Bridge Based on Multiple Transformers

**DOI:** 10.3390/s24123844

**Published:** 2024-06-14

**Authors:** Xin Liu, Yuzhu Chen, Longqi Wang, Tao Yu, Zhi Wang, Ke Xue, Yanlin Sui, Yongkun Chen

**Affiliations:** Changchun Institute of Optics, Fine Mechanics and Physics, Chinese Academy of Sciences, Changchun 130033, China; liuxciomp@163.com (X.L.); chenyz_mail@163.com (Y.C.); yut@ciomp.ac.cn (T.Y.); xuekeciomp@163.com (K.X.); suiyanlin@ciomp.ac.cn (Y.S.); ykchenscut@163.com (Y.C.)

**Keywords:** gravitational wave, resonance bridge, transformer, low-frequency measurement, noise analysis

## Abstract

The Taiji program is dedicated to the detection of middle and low-frequency gravitational waves, targeting the 0.1 mHz to 1 Hz frequency band. The project requires an acceleration residual sensitivity of 3 × 10^−15^ ms^−2^/Hz^1/2^, which necessitates a capacitance sensing resolution of 1 aF/Hz^1/2^ for the capacitive sensing system within the specified frequency range. The noise level of the resonant bridge significantly influences the resolution. Addressing the challenges in enhancing transformer performance parameters in existing resonant capacitance bridges and the constraints on improving the characteristics of resonant capacitance bridges, this study introduces a novel approach to reduce bridge thermal noise without optimizing existing parameters. The simulation results demonstrate that this scheme can reduce the noise to 0.7 times the original level and further reduce bridge thermal noise when other parameters affecting noise are optimized. This not only mitigates the demands for other performance parameters but also increases the range of maximum acceptable resonant frequency deviations and reduces its sensitivity to such variations. Experimental validation confirms that the proposed scheme effectively reduces noise by 0.7 times and improves the resolution of capacitance sensing to 0.6 aF/Hz^1/2^, thereby advancing the Taiji program gravitational wave detection capabilities.

## 1. Introduction

The detection of gravitational waves has long been a pursuit of scientists, serving not only to corroborate the predictions of general relativity but also to charter a new frontier in astronomy—gravitational wave astrophysics. This developing area has the potential to introduce a novel perspective on the universe, complementing our understanding gleaned from the electromagnetic spectrum [1]. Several projects have been initiated for the detection of gravitational waves. Notably, the Laser Interferometer Space Antenna (LISA), coordinated by the European Space Agency (ESA), is focused on the detection of gravitational waves within the frequency range from 0.1 mHz to 0.1 Hz, which are expected to arise from catastrophic events across the universe [2,3,4]. The TianQin project, initiated by Sun Yat-sen University, is tasked with conducting space-based gravitational wave detection within the frequency band range from 0.1 mHz to 1 Hz from a geocentric orbit [5,6,7]. The Taiji program, initiated by the Chinese Academy of Sciences, is a space mission dedicated to the detection of gravitational waves within the frequency band range from 0.1 mHz to 1 Hz. The primary objective of this mission is to measure the coalescence of supermassive (intermediate-mass) black hole binaries and the inspiral of extreme (intermediate-mass) ratio binaries [8,9,10,11].

Inertial sensors are critical components in the detection of gravitational waves, and due to their high-precision demands, the majority of research efforts currently utilize capacitive displacement sensors. The simplified system architecture is illustrated in Figure 1, which primarily comprises a test mass (simulator), a resonant capacitive bridge, a set of transimpedance amplifier modules, a set of AC amplification modules, a bandpass filtering module, a demodulation module, a data acquisition module, an injection voltage module, and digital circuitry.

The majority of current studies on the precision of capacitive displacement sensors primarily concentrate on noise reduction techniques, which are employed to enhance the detection accuracy of the sensors. The research subjects typically pertain to the individual modules represented in Figure 1.

Luigi Ferraioli has posited that within the frequency band range from 0.1 mHz to 1 Hz, the signal is predominantly governed by the force noise affecting the test mass. The investigator has conducted an extensive analysis of the expected statistical properties of the spectral data and has proposed two adequate methods for the estimation of excess noise [12]. Frank Steier conducted an experimental study to evaluate the influence of residual test quality angular noise on the longitudinal test quality displacement noise. Employing the linear least squares method, he determined the coupling factor for the angular fluctuations [13]. Ke Li analyzed the influence of the amplitude stability of the AC excitation signal on the low-frequency performance and dynamic range of inertial sensors. He identified the factors affecting amplitude stability and implemented a stable AC excitation signal [14]. Yuzhu Chen investigated the impact of the capacitive sensing fraction on the capacitive sensing resolution. He compared the performance of transformers with different quality factors, analyzed the factors influencing the transimpedance amplifier (TIA) noise, and examined the factors affecting the stability of the excitation signal [15]. Ke Xue examined the influence of the phase alignment of the demodulation switch signal in a synchronous demodulator on the capacitive sensing voltage noise. He proposed a method for achieving phase alignment [16]. Chengrui Wang implemented a capacitive sensing scheme based on a transformer bridge and synchronous demodulation, enhancing the functionality of the phase-locked loop (PLL) amplifier circuit and reducing the low-frequency 1/f noise [17]. Ke Li conducted an analysis on the differential capacitive displacement sensor, focusing on how the stability of the carrier signal and the phase difference and fluctuations between the demodulation signals can restrict the modulation–demodulation process and reduce capacitive sensitivity. He proposed a novel modulation–demodulation scheme based on a step-sine carrier wave, which can effectively mitigate these impacts [18]. Jianbo Yu investigated the influence of coaxial cables employed in inertial sensors on stray capacitance. He analyzed the measurement model of the coaxial cables and the noise model incorporating stray capacitance, exploring the effects of the cables. Finally, he conducted experimental measurements to assess the noise level of the cables at 0.1 Hz [19]. In the context of weak signal detection, Fang Yixi proposed a method in which the combination of hardware compensation and the corresponding algorithm cancellation are used to overcome the noise drifts and improve the detection accuracy of the high-precision temperature measurement system [20].

In the context of our research initiatives, it is imperative to address the amplification of front-end noise by subsequent amplification circuitry. Consequently, the attenuation of noise within the front-end resonance capacitor bridge circuit assumes critical importance. The current research results indicate that the primary factors contributing to the noise signature are encapsulated in Equation (1), which provides a quantitative representation of these influences [1].
(1)$$S_{\Delta C}^{1/2}=\frac{1}{U_{M}}\sqrt{\frac{{4k}_{B}T}{\omega_{0}^{3}\mathrm{LQ}}}$$
where $U_{M}$ is the injection voltage at the TM (test mass), $k_{B}$ is the Boltzmann constant, T is temperature, $\omega_{0}$ is the angular velocity at the resonant frequency, L is the inductance of the transformer at the resonant frequency, and Q is the quality factor of the transformer at the resonant frequency.

The prevalent research in the field of noise reduction primarily aims to improve the performance of the parameters within the formula in order to mitigate the noise generated by resonant capacitor bridges. Yafei Xie has modeled and analyzed the transfer function and capacitive noise of capacitive sensing technology circuits. Xie suggests that the product of L and Q and the enhancement of the resonant frequency are critical factors for improving the resolution of capacitive measurement. Xie proposed a graphical representation of the product of L and Q, and at a frequency of 200 kHz, he measured the capacitive sensing resolution [20]. S. Saraf has introduced a design for a precise AC resonant bridge sensor and analyzed the effects of transformer core gaps and temperature on sensitivity. The results indicate that a magnetic core without the opening exhibits a better match with the system, and lower temperatures can further reduce noise levels [21]. Mance has also concluded that increasing the transformer inductance (L) results in a shift in the frequency at which the maximum quality factor (Q) is achieved, which in turn leads to a reduction in the quality factor at the resonant frequency [1]. This also suggests that although increasing the transformer inductance can reduce noise, there are limitations to raising the inductance value. When the product of L and Q reaches a maximum, it becomes challenging to further enhance the value. M. Hu examined the front-end electronics of an LC resonant bridge sensor and investigated the influence of AC excitation signal frequency misalignment with the actual resonant frequency on output voltage noise, proposing methods for calibration. The noise characteristics of the capacitive sensor’s front-end circuit were analyzed, and both the sensitivity and noise models were refined. Utilizing the updated model, the total stray capacitance was optimized, leading to a reduction in noise levels [22]. Yanlin Sui has suggested that the primary factors contributing to the noise in the low-frequency band of gravitational wave detection are device thermal noise and 1/f noise. He has implemented a low-frequency, high-precision resonant capacitive bridge utilizing planar transformers with low temperature drift and low 1/f noise properties. Furthermore, Sui conducted closed-loop tests of the capacitive sensing circuit and the sensitive structure. The capacitive sensing noise within the 10 mHz–1 Hz frequency band achieved a level of 1.095 aF/Hz^1/2^ [23].

Prior to the initiation of the present study, the research team had successively accomplished a series of foundational investigations. These included the stability assessment of the injection voltage, the development of a low-noise test quality simulator, the analysis of noise within the transimpedance amplifier (TIA), the examination of noise in the demodulation module, and the exploration of the impacts of resonant frequency shifts. Following a detailed analysis of the proportional contributions of noise from various segments, it was identified that bridge noise is the primary factor currently influencing system performance. Contrary to approaches that focus on enhancing the performance of individual parameters, this study introduces a high-performance resonant bridge scheme that does not necessitate improvements in existing parameter performances, yet significantly reduces noise and enhances the capacitive sensing resolution. This solution aims to address the current limitations in enhancing transformer parameters, as well as the saturation of bridge thermal noise.

The present study first delineates the detailed implementation scheme of a high-precision resonant bridge based on a dual-transformer configuration. The simplified system is subsequently modeled, and the resonant conditions, primary noise components, and their influencing parameters are analyzed. Based on this foundation, simulations utilizing transformers with varying parameters are conducted. The proposed scheme in this paper is found to reduce the bridge noise by approximately 0.71 times compared to the original level.

## 2. Derivation in the Resonant Bridge Model and Noise Analysis

### 2.1. Resonant Capacitive Bridge Model

The presented scheme is a resonant capacitive bridge based on a dual-transformer configuration, whose bridge configuration and the cross-connected manner between transformers are illustrated in Figure 2.

In the scheme described, $C_{t1}$ and $C_{t2}$ represent the tuning capacitors, and $C_{a1}$ and $C_{a2}$ are allocated as drive-function reserved capacitors, which can also be utilized to adjust the symmetry [24]. The transformer pair is composed of transformer 1 ($T_{1}$) and transformer 2 ($T_{2}$). $L_{A}$, $L_{B}$, and $L_{C}$ denote the primary and secondary windings of transformer 1, while $L_{a}$, $L_{b}$, and $L_{c}$ signify the primary and secondary windings of transformer 2. $C_{j1}$ and $C_{j2}$ serve as the decoupling capacitors between the transformers and the transimpedance amplifier (TIA). $R_{f1}$ and $R_{f2}$ are the feedback resistors, while $C_{f1}$ and $C_{f2}$ are the feedback capacitors. For the sake of analyzing system performance, Figure 2 is simplified to the circuit representation in Figure 3.

The current through the primary winding of the transformer, I_p1_ and I_p2_, has been solved using Kirchhoff’s circuit laws.
(2)$$I_{p1}={\mathrm{sC}}_{1}\left(U_{M}-U_{1}\right)-{\mathrm{sC}}_{t1}U_{1}={\mathrm{sC}}_{1}U_{M}-s\left(C_{1}+C_{t1}\right)U_{1}$$
(3)$$I_{p2}={\mathrm{sC}}_{2}\left(U_{M}-U_{2}\right)-{\mathrm{sC}}_{t2}U_{2}={\mathrm{sC}}_{2}U_{M}-s\left(C_{2}+C_{t2}\right)U_{2}$$
where $U_{1}=U_{A}+U_{a}$ and $U_{2}=U_{B}+U_{b}$.

The number of turns in the transformer windings being equal allows for the derivation of the following relationship: $U_{1}=-U_{2}=U_{S}$.
(4)$$U_{s1}={\mathrm{sL}}_{C}I_{s}+{\mathrm{sM}}_{\mathrm{AC}}I_{p1}-{\mathrm{sM}}_{\mathrm{BC}}I_{p2}$$
(5)$$U_{s2}={\mathrm{sL}}_{c}I_{s}+{\mathrm{sM}}_{\mathrm{ac}}I_{p1}-{\mathrm{sM}}_{\mathrm{bc}}I_{p2}$$
where $M_{\mathrm{AC}}=K_{11}\sqrt{L_{A}L_{C}}$, $M_{\mathrm{BC}}=K_{12}\sqrt{L_{B}L_{C}}$, $M_{\mathrm{ac}}=K_{21}\sqrt{L_{a}L_{c}}$, and $M_{\mathrm{bc}}=K_{22}\sqrt{L_{b}L_{c}}$. The coupling coefficients for the windings in the transformers are denoted as follows: K_11_ corresponds to the coupling coefficient between windings A and C in transformer 1, K_12_ represents the coupling coefficient between windings B and C in transformer 1, K_21_ indicates the coupling coefficient between windings a and c in transformer 2, and K_22_ signifies the coupling coefficient between windings b and c in transformer 2.

Suppose an open-circuit condition arises at the terminal of the transformer’s secondary winding, $I_{s}=0$.
(6)$$U_{S}=s\left(M_{\mathrm{AC}}+M_{\mathrm{ac}}\right)I_{p1}-s\left(M_{\mathrm{BC}}+M_{\mathrm{bc}}\right)I_{p2}$$

The output voltage of the sensing bridge is represented by $U_{\mathrm{BR}}$. Equations (2) and (3) are incorporated into Equation (6):(7)$$U_{S}=U_{\mathrm{BR}}=\frac{s^{2}U_{M}\left[\left(M_{\mathrm{AC}}+M_{\mathrm{ac}}\right)C_{1}-\left(M_{\mathrm{BC}}+M_{\mathrm{bc}}\right)C_{2}\right]}{1+s^{2}\left[\left(M_{\mathrm{AC}}+M_{\mathrm{ac}}\right)\left(C_{1}+C_{t1}\right)+\left(M_{\mathrm{BC}}+M_{\mathrm{bc}}\right)\left(C_{2}+C_{t2}\right)\right]}.$$

Upon short-circuit occurrence at the terminals of the secondary circuit, $U_{S}=0$. Substituting this into Equations (4) and (5) yields the following:(8)$$I_{s}=\frac{-s\left(M_{\mathrm{AC}}+M_{\mathrm{ac}}\right)C_{1}U_{M}}{\left(L_{C}+L_{c}\right)}+\frac{s\left(M_{\mathrm{BC}}+M_{\mathrm{bc}}\right)C_{2}U_{M}}{\left(L_{C}+L_{c}\right)}.$$

The output impedance of the sensing bridge is represented by $Z_{\mathrm{BR}}$. By combining Equations (7) and (8), it can be derived that
(9)$$Z_{\mathrm{BR}}=\left|\frac{U_{\mathrm{BR}}}{I_{s}}\right|=\frac{s\left(L_{C}+L_{c}\right)}{1+s^{2}\left[\left(M_{\mathrm{AC}}+M_{\mathrm{ac}}\right)\left(C_{1}+C_{t1}\right)+\left(M_{\mathrm{BC}}+M_{\mathrm{bc}}\right)\left(C_{2}+C_{t2}\right)\right]}.$$

Accounting for the actual losses in a real transformer,
(10)$$L_{R}=L\left(1-j\delta \right),$$
where $\delta =\frac{1}{Q}$.

Herein, the following assumptions are made: $L_{A}=L_{B}=L_{C}=L_{1}$, $L_{a}=L_{b}=L_{c}=L_{2}$, $Q_{A}=Q_{B}=Q_{C}=Q_{1}=\frac{1}{\delta_{1}}$, $Q_{a}=Q_{b}=Q_{c}=Q_{2}=\frac{1}{\delta_{1}}$, and $K_{11}=K_{21}=K_{12}=K_{22}=1$.

Substituting the actual value of $L_{R}$ for L in Equation (7) yields the following:(11)$$U_{\mathrm{BR}}=\frac{-U_{M}\omega^{2}T_{\delta L}\left(C_{1}-C_{2}\right)}{1-\omega^{2}T_{\delta L}C_{\mathrm{eq}}},$$
where $C_{\mathrm{eq}}=C_{1}+C_{t1}+C_{2}+C_{t2}$ and $T_{\delta L}=\left(1-{j\delta }_{1}\right)L_{1}+\left(1-{j\delta }_{2}\right)L_{2}$.



(12)
$$I_{s}=-\Delta CU_{M}s$$


(13)
$$Z_{\mathrm{BR}}=\frac{j\omega \left[\left(1-{j\delta }_{1}\right)L_{1}+\left(1-{j\delta }_{2}\right)L_{2}\right]}{1-\omega^{2}\left[\left(1-{j\delta }_{1}\right)L_{1}C_{\mathrm{eq}}+\left(1-{j\delta }_{2}\right)L_{2}C_{\mathrm{eq}}\right]}$$



When the imaginary part of $Z_{\mathrm{BR}}$ is zero, the bridge is in a state of resonance.
(14)$$\omega =\sqrt{\frac{L_{1}+L_{2}}{C_{\mathrm{eq}}{\left(L_{1}+L_{2}\right)}^{2}+{\left(\frac{L_{1}}{Q_{1}}+\frac{L_{2}}{Q_{2}}\right)}^{2}}}$$

When the Q-value exceeds 100, the following approximation can be employed [1]:(15)$$\omega_{0}\approx \sqrt{\frac{1}{C_{\mathrm{eq}}\left(L_{1}+L_{2}\right)}}$$

$\omega_{0}$ represents the angular velocity at the resonant frequency.

It can be observed from Equation (15) that the resonant frequency is dependent on the tuning capacitance and the inductance values of the two transformers. At a fixed resonant frequency, the sum of the inductance values is inversely proportional to the tuning capacitance. Accordingly, as the inductance values increase, it leads to a reduction in the tuning capacitance $C_{\mathrm{eq}}$. The test mass, when connected to the bridge via coaxial cables, possesses a capacitance to ground, which contributes to the tuning capacitance. This implies that the size of the tuning capacitance determines the maximum length of the coaxial cables that can be utilized. A smaller tuning capacitance results in shorter maximum cable lengths. Addressing this issue, this study proposes two solutions: a passive bridge scheme and a low-capacitance cable scheme. However, these solutions are not the primary focus of this paper and thus will not be detailed here.

### 2.2. Alternative Connection Scheme

In addition to the aforementioned experimental protocol, this section will delineate a sequential cascade approach. Distinct from the earlier described scheme, the inter-winding connectivity is altered. The diagram illustrating the cross-connection method between transformers is provided in Figure 4.

Simulation validation has been performed, and the results suggest that the efficacy is comparable to that observed in Section 2.1 for Figure 2.

The primary winding $L_{A}$ of transformer 1 ($T_{1}$) is connected to the corresponding terminal of the primary winding $L_{B}$ of the same transformer, creating a novel primary winding configuration. Similarly, the primary winding $L_{a}$ of transformer 2 ($T_{2}$) is joined to the corresponding terminal of the primary winding $L_{b}$ within the same transformer, thereby forming another novel primary winding. The heterogeneous terminals of the secondary windings $L_{C}$ from transformer 1 ($T_{1}$) and the secondary winding $L_{c}$ from transformer 2 ($T_{2}$) are interconnected to establish a new secondary winding. This newly formed secondary winding is then terminated between two transimpedance amplifiers.

The simulation results have been obtained to validate the proposed configuration, and the findings indicate that it exhibits a performance equivalent to that of Figure 2. This scheme also offers insights for the development of subsequent transformer structures with novel configurations.

### 2.3. A Scheme for Lower Noise Generation

The previous sections have primarily introduced a resonant bridge scheme based on dual transformers. However, this scheme is not limited to two transformers. Theoretical analysis indicates that employing a greater number of transformers can achieve lower noise levels. Taking four transformers (T_a_, T_b_, T_c_, and T_d_) as an example, detailed deduction formulas are not provided in this paper, but a detailed implementation scheme is presented in the form of a connection diagram, as shown in Figure 5.

However, the adoption of a multi-transformer scheme may introduce two potential issues. The first is that an increased number of transformers will consume a larger physical space. The second is that, due to the overall tuning capacitance size, the maximum allowable length of the interconnecting cables will be constrained.

### 2.4. Noise Analysis and Simulation

The previous chapter delineated the operational principles and resonant conditions of the bridge circuit. This section will exemplify the dual-transformer scheme introduced in Section 2.1 to analyze its noise performance. To comprehensively analyze the impact of the novel scheme on the overall sensor system noise, it is imperative to not only consider the noise from the resonant bridge but to also take into account the noise generated by the transimpedance amplifier (TIA). The analysis is thus segmented into four primary components: the noise equivalent thermal impedance of the sensor bridge ($e_{\mathrm{BR}}$), the real-part thermal noise of the TIA feedback impedance ($e_{\mathrm{ZF}}$), the equivalent voltage noise resulting from the TIA current noise ($e_{\mathrm{in}}$), and the TIA voltage noise ($e_{\mathrm{un}}$) [1]. The real part of the complex bridge impedance is represented by $R_{Z_{\mathrm{BR}}}$.
(16)$$e_{\mathrm{BR}}=G_{\mathrm{TIA}}\sqrt{{4k}_{B}{\mathrm{TR}}_{Z_{\mathrm{BR}}}}$$
where $G_{\mathrm{TIA}}\approx \frac{2}{{\omega C}_{F}\left|Z_{\mathrm{BR}}\right|}$.
(17)$$e_{\mathrm{in}}={2i}_{\mathrm{AMP}}\left|Z_{\mathrm{FB}}\right|$$
(18)$$e_{\mathrm{un}}={2u}_{\mathrm{AMP}}\mathrm{NG}$$
where $\mathrm{NG}\approx 1+2\frac{Z_{\mathrm{FB}}}{Z_{\mathrm{BR}}}$. The real part of the complex TIA feedback impedance is represented by $R_{Z_{\mathrm{FB}}}$.
(19)$$e_{\mathrm{TH}}=G_{\mathrm{TIA}}\sqrt{{4k}_{B}{\mathrm{TR}}_{Z_{\mathrm{FB}}}}$$

To meet the noise requirement of 1 aF/Hz^1/2^ [1], it is necessary for the total voltage noise to satisfy 353 nV/Hz^1/2^.

Simulations were conducted with diverse parameters. The inductance values of the two transformers were similar, whereas their quality factors differed by nearly a factor of two. The effectiveness of the scheme in noise suppression when employing transformers with different parameters was compared.

The transformer parameters and circuit simulation parameters used in the simulations are presented in Table 1. In the context, T3 refers to the first type of transformer, while T4 denotes the second type. The original scheme is represented by “Original” and the newly proposed scheme is indicated as “New”.

The results of the Figure 6 and Figure 7 are summarized in Table 2.

The simulation results indicate that the proposed scheme achieves the minimum circuit noise at the resonant frequency. At this frequency, the other three types of noise are significantly lower than the $e_{\mathrm{BR}}$, making them the primary contributors to circuit noise. In the proposed scheme, the $e_{\mathrm{BR}}$ is reduced from 2.385 × 10^−7^ to 1.687 × 10^−7^, approximately 0.707 times the original value, and the $e_{\mathrm{un}}$ is decreased from 1.657 × 10^−8^ to 8.506 × 10^−9^, approximately 0.51 times the original value, at the resonant frequency. The real part of the TIA feedback impedance thermal noise $e_{\mathrm{ZF}}$ and the equivalent voltage noise $e_{\mathrm{in}}$ produced by the TIA current noise remain almost unchanged. Utilizing $T_{4}$ yields conclusions consistent with those from $T_{3}$. The results indicate that the proposed scheme primarily reduces $e_{\mathrm{BR}}$ and $e_{\mathrm{un}}$.

Upon deviation from the resonant frequency, the noise $e_{\mathrm{un}}$ increases. The simulation results suggest that when there is significant deviation from the resonant frequency, the impact of this deviation exceeds that of the $e_{\mathrm{BR}}$, becoming a predominant factor affecting noise performance. Consequently, to ensure the system’s noise performance, it is crucial to maintain the system’s operation in the vicinity of the resonant frequency.

In practical applications, considering only the results at the resonant frequency is incomplete due to the influence of stray capacitance and the experimental environment. Therefore, it is necessary to analyze the situation when the resonant frequency is offset. The system requires a noise level of 1 aF/Hz^1/2^, which necessitates the front-end circuit noise to be greater than 353 nV/Hz^1/2^.

The resonant frequency span for each scheme that meets the requirement of 1 aF/Hz^1/2^ in Figure 8 is summarized in Table 3.

Additionally, the graph illustrates that the noise curve is asymmetric with respect to the resonant frequency in linear coordinates. On the side where the resonant frequency decreases, the noise threshold is reached at frequency offsets of 3.63 kHz, 8.8 kHz, 4.5 kHz, and 8.9 kHz. Conversely, on the side where the resonant frequency increases, the noise threshold is reached at frequency offsets of 4.6 kHz, 12 kHz, 5.4 kHz, and 12.8 kHz. The side with the increasing resonant frequency exhibits a higher margin.

Furthermore, the novel scheme exhibits lower noise at the resonant frequency and higher sensor accuracy. Compared to the original scheme, the curve slope in the new scheme is gentler, indicating that for the same resonant frequency offset, the noise increment is minimal, making the scheme relatively insensitive to resonant frequency deviations. During application, variations in temperature, cable bending, and changes in vacuum levels can cause shifts in the resonant frequency. This suggests that the new scheme, leveraging its inherent advantages, can suppress the impact of resonant frequency shifts on noise and accuracy, thereby better adapting to experimental conditions.

The simulation results comparing transformers with varying parameters indicate that noise levels are reduced across the board when employing transformers with different parameters, with the noise being approximately 0.72 times that of the original scheme. This demonstrates the versatility of the scheme. Furthermore, when using transformers with inferior parameters, the noise is reduced by approximately 0.72 times, and when using transformers with superior parameters, the noise can still be reduced to about 0.72 times the original noise. This confirms that the scheme can further reduce noise to approximately 0.72 times the original noise even when implementing circuit performance enhancements such as reducing temperature, increasing the voltage of the injection voltage, and decreasing the noise of the injection voltage, which are strategies to mitigate system noise. This also suggests that the scheme can achieve a commendable noise level even with transformers of lesser performance, thereby reducing the system’s reliance on higher parameter performance.

## 3. Experimental Results

### 3.1. Optimization of Experimental Conditions and Parameters

During the testing process, to validate the effectiveness of the bridge scheme, planar transformers (P-T_1_/P-T_2_) and conventional transformers (C-T_1_/C-T_2_) with different parameters were utilized for separate tests. Comparative experiments were also conducted with the original scheme. The system composition is illustrated in Figure 9, which includes a test mass simulator, an FPGA, an injecting voltage module, a capacitive sensing system, and a data acquisition module.

The experimental environment is depicted in the Figure 10.

Figure 11 illustrates that, taking the planar transformer as an example, the red cable connects the like terminals of the windings, while the black cable connects the opposite terminals of the windings. The opposite terminal (black) of the primary winding L_A_ in transformer 1 is connected to the like terminal (red) of the primary winding L_a_ in transformer 2. The like terminal (red) of the primary winding L_A_ and the opposite terminal (black) of the primary winding L_a_ are collectively utilized as a novel primary winding to connect to the circuit. The opposite terminal (black) of the primary winding L_B_ in transformer 1 is connected to the like terminal (red) of the primary winding L_b_ in transformer 2. The like terminal (red) of the primary winding L_B_ and the opposite terminal (black) of the primary winding L_b_ are collectively utilized as a novel primary winding to connect to the circuit. The like terminal (red) of the secondary winding L_C_ in transformer 1 is connected to the opposite terminal (black) of the secondary winding L_c_ in transformer 2. The opposite terminal (black) of the secondary winding L_C_ and the like terminal (red) of the secondary winding L_c_ are collectively utilized as a novel secondary winding to connect to the circuit.

In Table 4, ‘P-T_1_’ and ‘P-T_2_’ refer to the transformers used in the planar transformer scheme, while ‘C-T_1_’ and ‘C-T_2_’ denote the transformers utilized in the conventional transformer scheme.

### 3.2. Planar Transformer Test Results

In the preliminary evaluation of tolerance and gain in the capacitance sensing system for quality testing, the tolerance was 4.3 fF and the gain was 40 V/pF.

The results of the voltage noise test are presented in Figure 12. The red curve in the figure represents the discrete test results, while the blue curve corresponds to the linearized test results.

The voltage noise test results using the planar transformers are depicted in the figure above. In the original scheme, the voltage noise was measured at 42.4 μV/Hz^1/2^@10 mHz, whereas in the new scheme, the voltage noise remained at 27.2 μV/Hz^1/2^@10 mHz. The voltage noise was reduced to 0.64 times that of the original scheme.

The blue curve represents the Linear Power Spectral Density (LPSD). At low frequencies, the power spectrum is calculated by reducing the sampling rate; at high frequencies, only a segment of the data is used, thereby reducing the data length and saving time. At high frequencies, the width of the lines is significantly improved. The frequency resolution is smaller at low frequencies and gradually increases at high frequencies. The red curve represents the Power Spectral Density (PSD). The periodogram method is employed for period extension of the random process x(t). The resolution at both high and low frequencies is the same, and the PSD (red curve) provides a more accurate description of the actual performance of the system.

The results of the capacitance noise test are presented in Figure 13.

Capacitive sensing noise tests were conducted using the planar transformers. When employing the original bridge scheme, the capacitive sensing noise measurement yielded a result of 1.06 aF/Hz^1/2^@10 mHz. With the implementation of the new scheme, the capacitive sensing noise was measured at 0.68 aF/Hz^1/2^@10 mHz. The new scheme reduced the noise level by approximately 0.64 times.

### 3.3. Conventional Transformer Test Results

In the preliminary evaluation of tolerance and gain in the capacitance sensing system for quality testing, the tolerance was 4.3 fF and the gain was 40 V/pF.

The results of the voltage noise test are presented in Figure 14. The red curve in the figure represents the discrete test results, while the blue curve corresponds to the linearized test results.

The voltage noise test results using the conventional transformers are depicted in the figure above. In the original scheme, the voltage noise was measured at 33.6 μV/Hz^1/2^@10 mHz, whereas in the new scheme, the voltage noise remained at 26 μV/Hz^1/2^@10 mHz. The voltage noise was reduced to 0.77 times that of the original scheme.

The results of the capacitance noise test are presented in Figure 15.

Capacitive sensing noise tests were conducted using the conventional transformers. When employing the original bridge scheme, the capacitive sensing noise measurement yielded a result of 0.84 aF/Hz^1/2^@10 mHz. With the implementation of the new scheme, the capacitive sensing noise was measured at 0.6 aF/Hz^1/2^@10 mHz. The new scheme reduced the noise level by approximately 0.71 times.

The above test results are summarized in the table below. In Table 5, ‘P-T’ represents the planar transformer scheme, while ‘C-T’ signifies the conventional transformer scheme.

Upon implementation of the new approach, the theoretical voltage noise limit was reduced to 0.7 times that of the original scheme, with the actual measured noise levels also dropping by approximately 0.64 times to 0.7 times. The current method for adjusting the resonant frequency involves observing whether the point of lowest noise is at the resonant frequency using a spectrum analyzer. Due to the impact of the resolution and the resolution bandwidth of the spectrum analyzer, there may be a certain degree of error in determining the resonant frequency, which in turn affects the results. Therefore, the actual noise reduction effects of the two transformers tested are not completely consistent, with values of 0.7 times and 0.64 times, respectively. Excluding this impact, the noise reduction effect should be consistent with the theoretical analysis result, around 0.7 times. The experimental results were found to be in substantial agreement with the theoretical expectations, thus validating the effectiveness of the proposed scheme. When compared to planar transformers and traditional transformers with better parameters, the noise was reduced by approximately 0.7 times. However, with the adoption of the scheme presented in this paper, the noise was further reduced by an additional 0.7 times, indicating that when transformer parameters are improved, the application of this scheme can still achieve a further reduction of 0.7 times based on the enhanced parameters. Moreover, when employing planar transformers with poorer parameters, the scheme can still approximate the performance of traditional transformers, relaxing the requirements for circuit parameters and providing favorable conditions for the application of planar transformers with good uniformity but inferior parameters.

However, it is also noted that under the new scheme, the tuned capacitance is reduced to half of its original value. The transmission line capacitance is a component of the tuned capacitance, and longer transmission lines result in greater capacitance. Therefore, the reduction in tuned capacitance will impact the length of the transmission lines. This could potentially lead to an insufficient transmission distance. In such a scenario, a passive bridge and low-capacitance cable scheme can be utilized, which has already been researched and has yielded certain results, although it is not the primary focus of this paper.

## 4. Conclusions

This paper investigates noise suppression in capacitive sensing systems, addressing the issue that the thermal noise of the resonant capacitance bridge has reached its limit, and the transformer parameters affecting the bridge’s thermal noise cannot be improved. We propose a high-precision resonant capacitance bridge scheme based on multiple transformers.

The simulation results with transformers of different parameters demonstrate that the proposed scheme reduces noise at the resonant frequency to 0.7 times its original level, thereby validating the effectiveness of the approach. Additionally, these results indicate that when other parameters are improved, the scheme can still further reduce the thermal noise of the resonant bridge, enhancing system performance. Furthermore, the scheme is not limited to noise reduction at the resonant frequency but is effective across a wide frequency band. The overall decrease in noise not only lowers the requirements for other performance parameters but also increases the margin of resonant frequency deviation for the system. Under the conditions of the new scheme, the system’s sensitivity to resonant frequency deviations is reduced, better meeting engineering application requirements.

The experimental results ultimately reveal that when utilizing traditional transformers, the voltage noise is reduced to approximately 0.7 times that of the original scheme, and the new approach can enhance the capacitive sensing resolution to 0.6 aF/Hz^1/2^. When employing planar transformers, the voltage noise is also reduced to about 0.7 times that of the original scheme, and the new approach can further improve the capacitive sensing resolution to 0.68 aF/Hz^1/2^. This indicates that the scheme can reduce the noise to approximately 0.64 times that of the original scheme, aligning closely with the results of the theoretical analysis. The potential issue of limited transmission distance is also discussed, and a solution involving a passive bridge and low-capacitance cable is provided, which has been preliminarily verified to effectively address the existing challenges.

Simultaneously, to fully exploit the performance of the scheme, there is a higher demand for accuracy in the detection and adjustment of the resonant frequency. Current methods involve observations with a frequency spectrum analyzer, which are limited by the instrument’s resolution and the variability in test results. Therefore, it is necessary to continue researching more precise resonant frequency measurement techniques to better utilize the system’s performance. Furthermore, during the selection of transformers, planar transformers exhibit better consistency but lower parameters. Developing planar transformers with better performance and different forms will also be the main research direction for subsequent work.

## Figures and Tables

**Figure 1 sensors-24-03844-f001:**
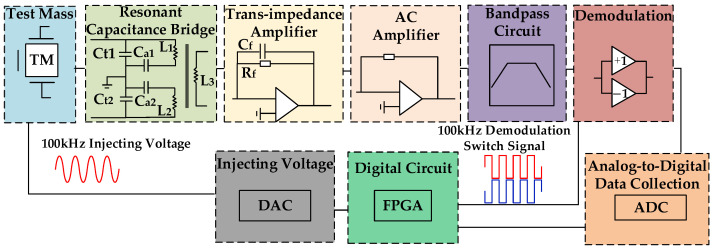
Schematic diagram of sensor system configuration.

**Figure 2 sensors-24-03844-f002:**
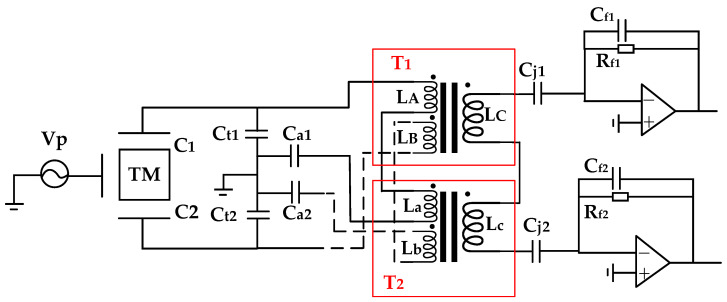
Simplified schematic diagram of scheme.

**Figure 3 sensors-24-03844-f003:**
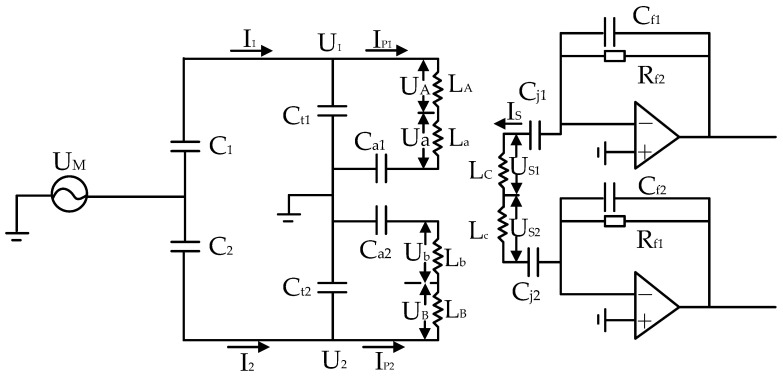
Simplified circuit model: transformers and TIAs.

**Figure 4 sensors-24-03844-f004:**
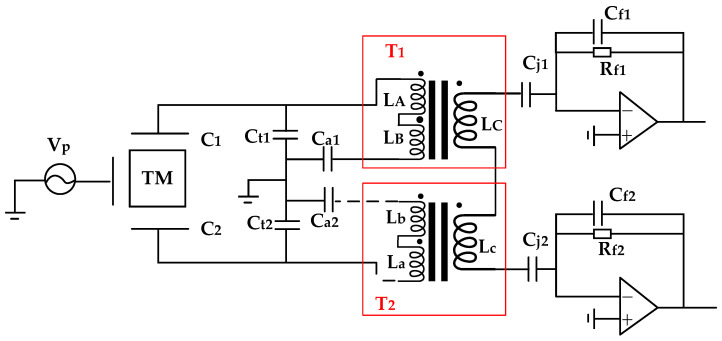
Simplified schematic diagram of cascaded scheme.

**Figure 5 sensors-24-03844-f005:**
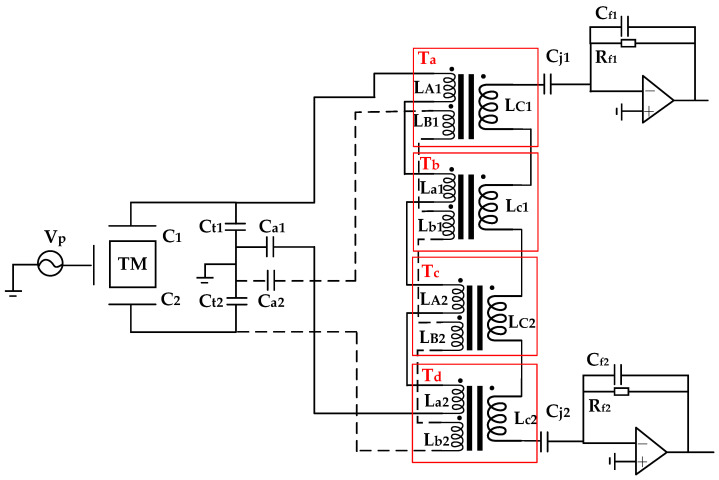
A diagram of a multi-transformer scheme.

**Figure 6 sensors-24-03844-f006:**
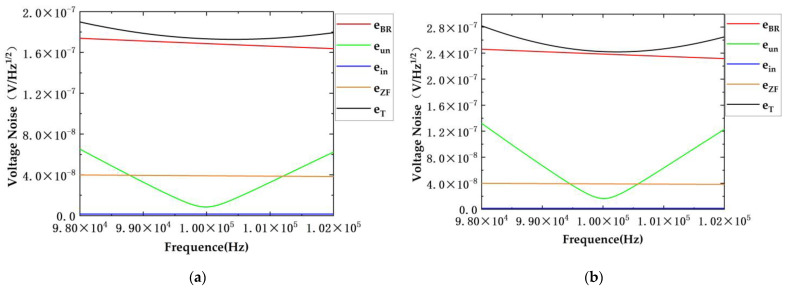
Simulation results of T3. (**a**) Simulation results of noise suppression in proposed dual-transformer scheme; (**b**) simulation results of noise suppression in original scheme.

**Figure 7 sensors-24-03844-f007:**
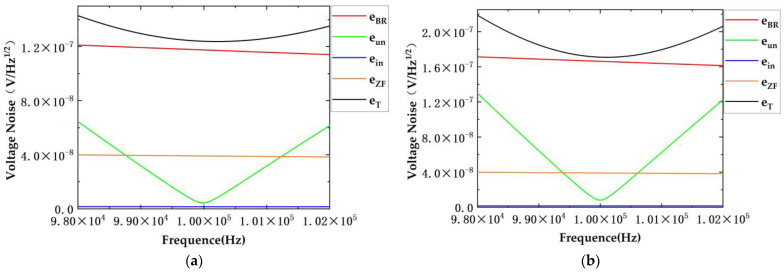
Simulation results of T4. (**a**) Simulation results of noise suppression in proposed dual-transformer scheme; (**b**) simulation results of noise suppression in original scheme.

**Figure 8 sensors-24-03844-f008:**
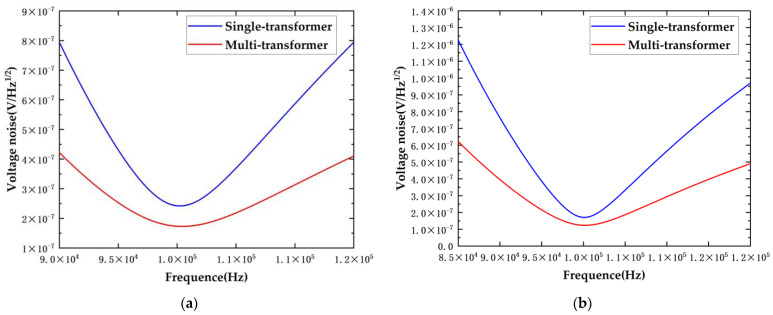
Comparative graph of wideband total noise for different schemes. (**a**) T3; (**b**) T4.

**Figure 9 sensors-24-03844-f009:**
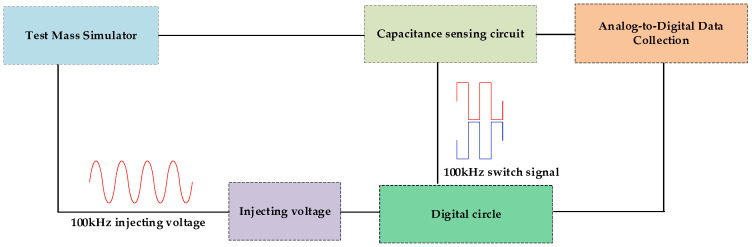
Schematic diagram of test system composition.

**Figure 10 sensors-24-03844-f010:**
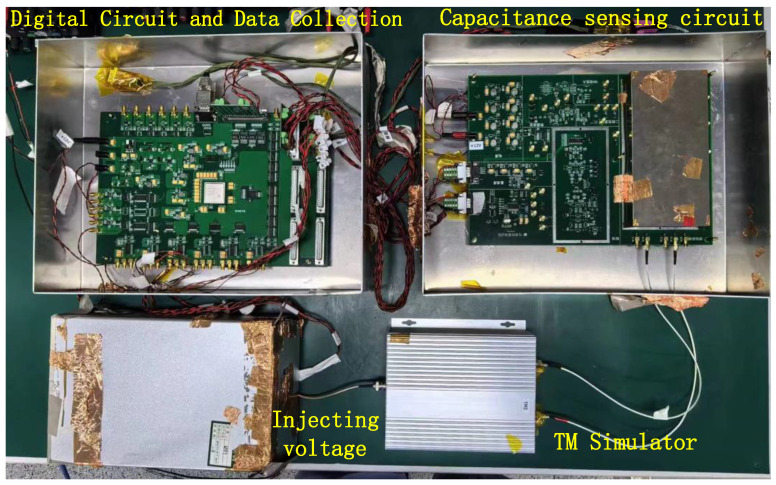
Testing environment.

**Figure 11 sensors-24-03844-f011:**
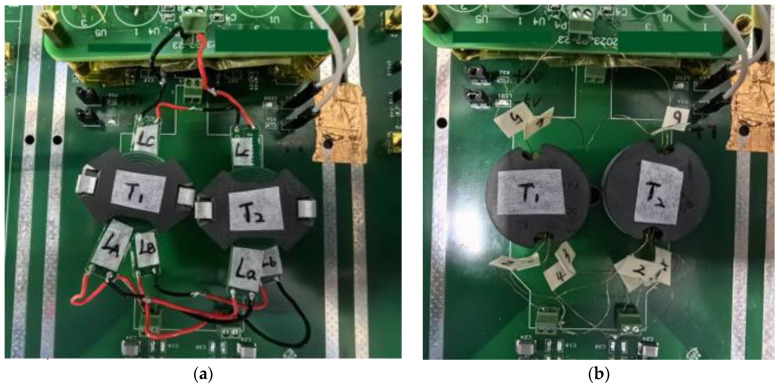
The actual connection diagrams of the planar transformers and the traditional transformers on the circuit board. (**a**) The planar transformers; (**b**) the traditional transformers.

**Figure 12 sensors-24-03844-f012:**
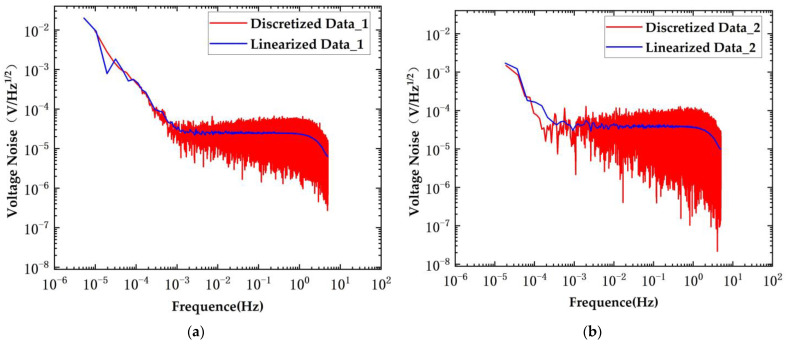
System voltage noise under conditions of planar transformers. (**a**) New scheme; (**b**) original scheme.

**Figure 13 sensors-24-03844-f013:**
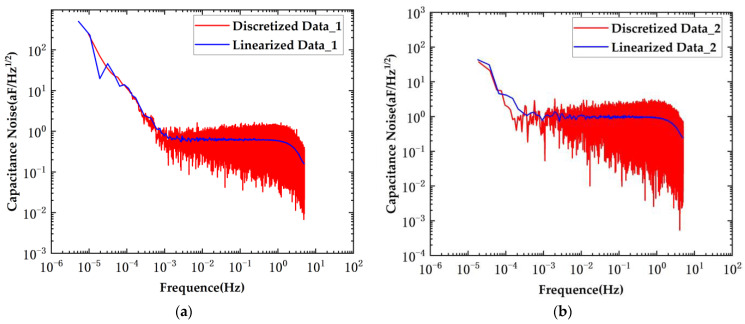
System capacitance noise under conditions of planar transformers. (**a**) New scheme; (**b**) original scheme.

**Figure 14 sensors-24-03844-f014:**
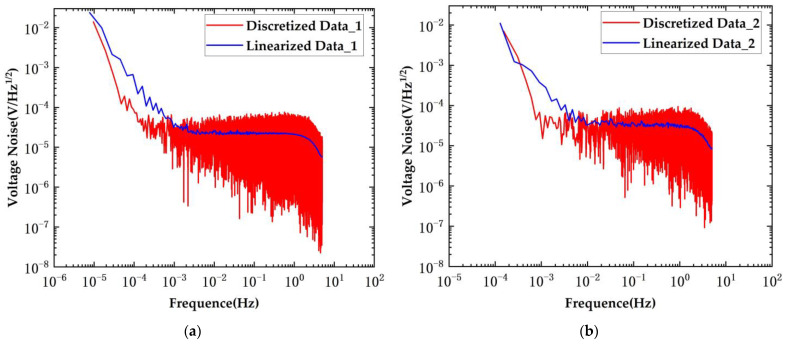
System voltage noise under conditions of conventional transformers. (**a**) New scheme; (**b**) original scheme.

**Figure 15 sensors-24-03844-f015:**
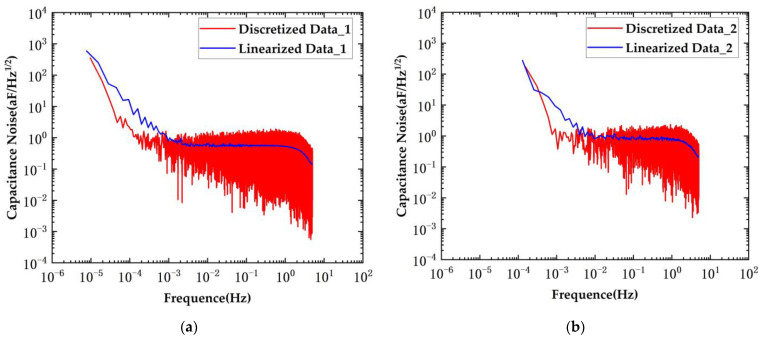
System capacitance noise under conditions of conventional transformers. (**a**) New scheme; (**b**) original scheme.

**Table 1 sensors-24-03844-t001:** Parameter table for simulations.

Parameters	$\mathrm{Original}\;(T_{3}$)	$\mathrm{New}\;(T_{3}$)	$\mathrm{Original}\;(T_{4}$)	New ($T_{4}$)
L	4.37 mH	4.37 mH	4.39 mH	4.39 mH
Q	196	196	402	402
K	0.97	0.97	0.97	0.97
$C_{\mathrm{eq}}$	304.68 pF	149.65 pF	304.5 pF	149.0 pF
$C_{q}$	611.56 pF	301.6 pF	611.2 pF	300.2 pF
$C_{0}$	1.15 pF	1.15 pF	1.15 pF	1.15 pF
$C_{j}$	3.3 nF	3.3 nF	3.3 nF	3.3 nF
$C_{F}$	3.3 pF	3.3 pF	3.3 pF	3.3 pF
$R_{F}$	10 MΩ	10 MΩ	10 MΩ	10 MΩ
$C_{a}$	10 nF	10 nF	10 nF	10 nF
$u_{\mathrm{AMP}}$	$14\;\mathrm{nV}/\sqrt{\mathrm{Hz}}$	$14\;\mathrm{nV}/\sqrt{\mathrm{Hz}}$	$14\;\mathrm{nV}/\sqrt{\mathrm{Hz}}$	$14\;\mathrm{nV}/\sqrt{\mathrm{Hz}}$
$i_{\mathrm{AMP}}$	$1.8\;\mathrm{fA}/\sqrt{\mathrm{Hz}}$	$1.8\;\mathrm{fA}/\sqrt{\mathrm{Hz}}$	$1.8\;\mathrm{fA}/\sqrt{\mathrm{Hz}}$	$1.8\;\mathrm{fA}/\sqrt{\mathrm{Hz}}$

**Table 2 sensors-24-03844-t002:** Noise table.

Scheme	$e_{\mathrm{BR}}$ (V/Hz^1/2^)	$e_{\mathrm{un}}$ (V/Hz^1/2^)	$e_{\mathrm{ZF}}$ (V/Hz^1/2^)	$e_{\mathrm{in}}$ (V/Hz^1/2^)	Total (V/Hz^1/2^)
Original ($T_{3}$)	2.385 × 10^−7^	1.657 × 10^−8^	3.909 × 10^−8^	1.542 × 10^−9^	2.423 × 10^−7^
New ($T_{3}$)	1.687 × 10^−7^	8.506 × 10^−9^	3.909 × 10^−8^	1.542 × 10^−9^	1.734 × 10^−7^
Original ($T_{4}$)	1.662 × 10^−7^	8.253 × 10^−9^	3.909 × 10^−8^	1.542 × 10^−9^	1.709 × 10^−7^
New ($T_{4}$)	1.175 × 10^−7^	4.361 × 10^−9^	3.909 × 10^−8^	1.542 × 10^−9^	1.239 × 10^−7^

**Table 3 sensors-24-03844-t003:** Frequency band statistical table.

Frequency	$\mathrm{Original}\;(T_{3}$)	$\mathrm{New}\;(T_{3}$)	$\mathrm{Original}\;(T_{4}$)	$\mathrm{New}\;(T_{4}$)
Starting frequency	96.37 kHz	91.8 kHz	95.5 kHz	91.1 kHz
Cutoff frequency	104.6 kHz	112 kHz	105.4 kHz	112.8 kHz
Frequency span	8.23 kHz	20.2 kHz	9.9 kHz	21.7 kHz

**Table 4 sensors-24-03844-t004:** Parameter table for experiments.

Parameters	P-T_1_	P-T_2_	C-T_1_	C-T_2_
L	4.37 mH	4.37 mH	4.39 mH	4.39 mH
Q	196	196	402	402
K	0.97	0.97	0.97	0.97
$C_{q}$	291 pF		285 pF	
$C_{0}$	1.1 pF		1.1 pF	
$C_{j}$	3.3 nF		3.3 nF	
$C_{F}$	3.3 pF		3.3 pF	
$R_{F}$	10 MΩ		10 MΩ	
$C_{a}$	10 nF		10 nF	
ΔC	4.3 fF		4.3 fF	

**Table 5 sensors-24-03844-t005:** Noise test results.

Parameters	Original (P-T)	New (P-T)	Original (C-T)	New (C-T)
Voltage noise (μV/Hz^1/2^@10 mHz)	42.4	27.2	33.6	26
Capacitance noise (aF/Hz^1/2^@10 mHz)	1.06	0.68	0.84	0.6

## Data Availability

The datasets presented in this article are not readily available because the data are part of an ongoing study or due to technical limitations. Requests to access the datasets should be directed to liuxciomp@163.com.

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
