# Peer review of "Research on High-Precision Resonant Capacitance Bridge Based on Multiple Transformers"

_sensors, 2024, doi:10.3390/s24123844_

Round 1

Reviewer 1 Report

Comments and Suggestions for Authors

The manuscript study on high-precision resonant capacitance bridge based on multiple transformers. The manuscript is of some significance. The recommendations are as follows.

1 Please summarize advantages and disadvantages of research on multiple transformers compared to traditional works.

2 The actual connection diagrams of the planar transformer and the traditional transformer on the circuit board are not shown. The differences between the actual connection and the ideal model are not analyzed in detail. Please add relevant pictures and descriptions.

3 It is stated that the planar transformer reduces noise slightly better than the conventional transformer, but the reasons for this are not explained.

4 Figures 11 to 14 present discrete test results (red curve) and linearized test results (blue curve). What do linearization and discretization mean in this context? Please add a description. There is no explanation for why the blue curve does not overlap with the red curve below 0.01 Hz. Is this discrepancy due to limitations in the linearization method?

5 Section 2.1 is titled "Noise Analysis," but it does not address the noise issue adequately. Instead, a noise analysis is given in Section 2.4. Please revise the chapter arrangement of Chapter 2.

Comments on the Quality of English Language

Minor editing of English language required

Author Response

We sincerely thank the Editor and the Reviewer for their valuable feedbacks and suggestions regarding the submitted paper.Please see the attachment.

Reviewer 2 Report

Comments and Suggestions for Authors

Review for the paper sensors-3026515

Research on High-Precision Resonant Capacitance Bridge Based on Multiple Transformers

Recommendations to improve:

1)      The abbreviation TM, which appears in line 95, is not defined in the text.

2)      Transformers T1 and T2 are not indicated in figure 2.

3)      The decoupling capacitors Cj1 and Cj2 are not given in figure 2.

4)      The source of formulae 2 and 3 is not clear.

5)      It is not clear what is s in formulae 2 and 3. If it s=jω, then it should be in the denominator.

6)      It is not clear what is k11, k12, k21 and k22 in line 168.

Conclusion. The paper needs revision according to the list above. 

Comments on the Quality of English Language

The language is good enough

Author Response

(The authors gave the same response as above.)

Reviewer 3 Report

Comments and Suggestions for Authors

This paper investigates noise suppression in capacitive sensing systems, addressing the problem that the thermal noise of the resonant capacitance bridge has reached its limit, and the transformer parameters that affect the thermal noise of the bridge cannot be improved.

The authors propose a high precision resonant capacitance bridge scheme based on multiple transformers. 

The simulation results with transformers with different parameters demonstrate the effectiveness of the proposed scheme to reduce the noise at the resonance frequency to 0.7 times its original level.

The work is well structured, the references are consistent. The proposed method is well argued and described and the results obtained support its performance.

Please redo the figures because they are not legible: 1,2, 5 (attention to the format), 6 (the legend is not legible).

Author Response

(The authors gave the same response as above.)

Round 2

Reviewer 1 Report

Comments and Suggestions for Authors

Accept in present form